# Impact of Annealing Temperature on the Morphological, Optical and Photoelectrochemical Properties of Cauliflower-like CdSe_0.6_Te_0.4_ Photoelectrodes; Enhanced Solar Cell Performance

**DOI:** 10.3390/ijms222111610

**Published:** 2021-10-27

**Authors:** Gajanan S. Ghodake, Dae-Young Kim, Surendra K. Shinde, Deepak P. Dubal, Hemraj M. Yadav, Verjesh Kumar Magotra

**Affiliations:** 1Department of Biological and Environmental Science, College of Life Science and Biotechnology, Dongguk University, 32 Dongguk-ro, Biomedical Campus, Ilsandong-gu, Siksa-dong, Goyang-si 10326, Korea; ghodakegs@gmail.com (G.S.G.); sbpkim@dongguk.edu (D.-Y.K.); hemrajy@gmail.com (H.M.Y.); 2School of Chemistry and Physics, Queensland University of Technology (QUT), 2 George Street, Brisbane, QLD 4001, Australia; deepak.dubal@qut.edu.au; 3Nano Information Technology Academy, Dongguk University, Seoul 100715, Korea; birju.srm@gmail.com

**Keywords:** electrodeposition, CdSe_0.6_Te_0.4_, thin films, XRD, EDS, DSSC, solar cell

## Abstract

We are reporting on the impact of air annealing temperatures on the physicochemical properties of electrochemically synthesized cadmium selenium telluride (CdSe_0.6_Te_0.4_) samples for their application in a photoelectrochemical (PEC) solar cell. The CdSe_0.6_Te_0.4_ samples were characterized with several sophisticated techniques to understand their characteristic properties. The XRD results presented the pure phase formation of the ternary CdSe_0.6_Te_0.4_ nanocompound with a hexagonal crystal structure, indicating that the annealing temperature influences the XRD peak intensity. The XPS study confirmed the existence of Cd, Se, and Te elements, indicating the formation of ternary CdSe_0.6_Te_0.4_ compounds. The FE-SEM results showed that the morphological engineering of the CdSe_0.6_Te_0.4_ samples can be achieved simply by changing the annealing temperatures from 300 to 400 °C with intervals of 50 °C. The efficiencies (ƞ) of the CdSe_0.6_Te_0.4_ photoelectrodes were found to be 2.0% for the non-annealed and 3.1, 3.6, and 2.5% for the annealed at 300, 350, and 400 °C, respectively. Most interestingly, the PEC cell analysis indicated that the annealing temperatures played an important role in boosting the performance of the photoelectrochemical properties of the solar cells.

## 1. Introduction

With the rapid increase in the world’s population and environmental pollution, there is limited availability of energy sources. A worldwide demand therefore exists for new energy sources that are clean, cost-effective, and simple and that do not cause environmental pollution [1,2]. Various energy storage devices available on the electronic markets include fuel cells [3], solar cells [4,5], H_2_ evolution systems [6,7], light-emitting diodes (LED) [8], capacitors [9], supercapacitors [10,11], and batteries [12]. Among these, the photoelectrochemical (PEC) cell is the best electronic device because of the easy and whole-day availability of sunlight on Earth, whereas the convenience of the other sources of materials is geographically limited. To improve the electrical properties of its solar cells, the PEC has two main principal requirements, which are related to the band gap energy and the stability of the photoelectrodes. First, the photoelectrode should provide the band gap of materials whose band gap energy is nearly matched to the extreme sunlight intensity in the visible spectrum to use the solar spectrum resourcefully (1–3 eV). Second, the semiconductor photoelectrode materials need to be stable with respect to the 1 M polysulfide (NaOH:Na_2_S:S) liquid electrolytes during the PEC measurements. 

To date, several binary metal chalcogenide photoelectrode materials have been developed into PECs as photoabsorber layers, including CdS [13,14], CdSe [15,16], CdTe [17], ZnS [18], ZnSe [19], and ZnSe/CdSe [20]. Among these binary photoelectrode materials, CdSe and CdTe materials provided band gap values between 1.45 and 1.75 eV [1,21,22,23,24,25,26,27,28,29,30]; this indicates that the lower band gap energy photoelectrodes provided higher optical absorption. The higher optical absorption of the photoelectrode material provides more electron–hole pairs as well as a higher rate of the recombination rates of charge carriers [21,22]. The significance of selecting particular compositions is that the CdSe_0.6_Te_0.4_ electrode shows a sharp transition from the properties of CdSe to those typical of CdTe in the composition [31]. The second reason for the selection of the ternary CdSe_0.6_Te_0.4_ compounds is to improve the electrical and optical properties of the binary CdSe and CdTe metal chalcogenides. Recently, various co-workers have been working on the CdSe_X_Te_1−X_ compounds and finally concluded that the CdSe_0.6_Te_0.4_ composition shows better photoelectrochemical properties than the other composition/photoelectrodes [23,24,25,26,27,28,29,30,31].

Currently, researchers have been using various chemical and physical deposition methods, such as molecular beam epitaxy (MBE) [23,24], chemical vapor [25], hydrothermal [26], radiofrequency sputtering [27], slurry coating [28], non-injection one-pot [29], silicon isolation layer [30], electrodeposition [31], and chemical bath deposition (CBD) [32], to develop ternary CdSe_0.6_Te_0.4_ thin films for various applications. Among these methods, the electrodeposition method is simple, rapid, and scalable for preparing ternary CdSe_0.6_Te_0.4_ samples [31,32,33,34,35,36,37]. Our previous study investigated the deposition of ternary CdSe_0.6_Te_0.4_ composites under different conditions, such as varying bath concentrations, deposition times, and electron-beam irradiation doses [31,32,33,34,35,36,37]. However, to the best of our knowledge, the effect of the annealing temperature at different temperature intervals of 50 °C on the morphological engineering of the ternary CdSe_0.6_Te_0.4_ composites for PEC has not been reported. 

In this work, we present the effect of different annealing temperatures on the structural, morphological, and photoelectrical properties of as-deposited CdSe_0.6_Te_0.4_ composites. As-fabricated CdSe_0.6_Te_0.4_ composites were subsequently characterized using physical techniques such as a structural analysis by XRD, compositional and elemental analyses by XPS and EDS, surface morphology observations by FE-SEM, and *J*-*V* measurements to evaluate their PEC properties. After the air annealing treatment, the CdSe_0.6_Te_0.4_ samples showed improvements in their structural, morphological, and photoelectrical properties. A maximum power efficiency (ƞ) of 3.6% for the CdSe_0.6_Te_0.4_ photoelectrode was realized when the sample was annealed at 350 °C at a 30 mV cm^−2^ power input in a 1 M polysulfide electrolyte. The FE-SEM results of the CdSe_0.6_Te_0.4_ samples show that variations in the annealing temperature result in different surface morphologies, which range from cauliflower-like to broccoli-like to coral-like nanostructures. 

## 2. Results and Discussion

### 2.1. X-ray Diffraction 

The non-annealed and annealed CdSe_0.6_Te_0.4_ composites were studied by XRD to identify their phase formation and crystal structures. Figure 1a shows the XRD patterns of the non-annealed CdSe_0.6_Te_0.4_ composites and the annealed at different temperatures. The peaks in the patterns obtained for all the ternary CdSe_0.6_Te_0.4_ composites are well matched to the standard JCPDS card number 41–1325. The peaks at 24.09°, 25.16°, 27.63°, 38.53°, 40.26°, 42.01°, 44.51°, 49.91°, 57.02°, and 63.69° correspond to the (100), (002), (101), (102), (110), (111), (103) (201), (202), and (203) diffraction planes, respectively, confirming the polycrystalline hexagonal crystal structure of both CdSe_0.6_Te_0.4_ samples [35,36,37]. After the annealing treatments, the XRD patterns of the CdSe_0.6_Te_0.4_ samples show increased sharpness and higher peak intensity as compared to the non-annealed samples. We clearly observed the CdSe_0.6_Te_0.4_ samples annealed at 350 °C, and the samples showed enhanced peak intensity and sharpness compared to all the CdSe_0.6_Te_0.4_ samples [38,39]. However, in the CdSe_0.6_Te_0.4_ samples annealed at the higher temperature of 400 °C, the XRD patterns show that the peak intensity decreases, confirming that the annealing temperature affects the crystallinity and peak sharpness of the CdSe_0.6_Te_0.4_ samples. The XRD results indicate the formation of the pure phase of the ternary CdSe_0.6_Te_0.4_ composites without any other impurities such as binary CdSe and CdTe nanomaterial. Su et al. [40] reported on CdSeTe quantum dots with a cubic crystal structure prepared via an aqueous solution method and their suitability for use in PECs and water-splitting applications; they concluded that the as-prepared CdSeTe quantum dots were useful for enhancing the photo power efficiency and stability of the solar cells. MacDonald et al. [41] explored the band gap energy and electrical properties of CdSe_x_Te_1−x_ nanocrystals with various CdSe and CdTe contents. We calculated the crystallite size of the non-annealed and annealed CdSe_0.6_Te_0.4_ samples at their greatest peak intensity (101) of the XRD patterns using standard Scherrer’s formula. The calculated crystallite size of the non-annealed samples was 36 nm; after annealing at 300, 350, and 400 °C, the crystallite sizes of the CdSe_0.6_Te_0.4_ samples were 27, 16, and 23 nm, respectively. The decrease in crystallite size from 36 to 16 nm can be attributed to the recrystallization of the CdSe_0.6_Te_0.4_ thin films at various temperatures. The obtained values of the crystallite sizes show that when annealed at a temperature of 350 °C, the crystallite size is smaller than the other samples, and this size contraction is attributable to the heat treatment [37]. 

### 2.2. X-ray Photoelectron Spectroscopy

Appendix A and Figure 1b show the survey spectra of the non-annealed CdSe_0.6_Te_0.4_ samples and the annealed at 350 °C to determine their elemental compositions and chemical valence states. Both spectra of samples show peaks associated with Cd 3d, Se 3d, C 1s, O 1s, and Te 3d elements, indicating the successful formation of ternary CdSe_0.6_Te_0.4_ thin films. Figure 1c–e shows the core-level spectra of Cd 3d, Se 3d, and Te 3d, respectively. As shown in Figure 1c, the major characteristic peaks at binding energies of 405.26 and 411.99 eV correspond to the Cd 3d_5/2_ and Cd 3d_3/2_ states, respectively [34]. The difference between the binding energies is approximately 6.73 eV, which is consistent with the formation of the ternary CdSe_0.6_Te_0.4_ compound [38]. The peaks observed at the binding energy of 400 eV, which is related to the surface of the Cd^2+^ ions, formed on the surface of the CdSe_0.6_Te_0.4_ samples [6,39,40]. The peak observed at 54.55 eV corresponds to the Se 3d region shown in Figure 1d, which indicates that the Se element is present in CdSe_0.6_Te_0.4_ sample. Figure 1e shows the Te 3d spectral region of the CdSe_0.6_Te_0.4_ sample annealed at a temperature of 350 °C. The spectra of the Te 3d show characteristic peaks at binding energies of 573.75 and 584.20 eV, which correspond to the Te 3d_5/2_ and Te 3d_3/2_ states, respectively (shown in Figure 1e) [4,40,41]. The Te 3d core-level spectra show the two other peaks observed at binding energies of 576.66 and 586.96 eV, which are assigned to Te 3d_5/2_ oxide and Te 3d_3/2_ oxide, respectively [42]. The XPS survey spectrum of both CdSe_0.6_Te_0.4_ samples shows the two other peaks at the binding energies of 284.56 and 531.56 for the C ls and O 1s peaks, respectively. The carbon peaks observed on the surface of the samples may be due to the carbon tape used for the preparation of the XPS analysis. The O 1s peaks can be attributed to the samples prepared in an aqueous solution and annealing in air atmospheric conditions. Thus, the XPS results confirm that Cd, Se, and Te elements are present in the CdSe_0.6_Te_0.4_ sample both before and after annealing, which indicates the formation of the CdSe_0.6_Te_0.4_ compound [43]. 

### 2.3. FT-IR

Figure 1f shows the FT-IR spectra of the non-annealed CdSe_0.6_Te_0.4_ and of the sample annealed at 350 °C to determine their vibrational characteristics. As shown in Figure 1f, the FT-IR spectrum of the non-annealed sample shows a broad characteristic peak at 3315 cm^−1^, which is related to the surface hydroxide (O–H) groups because of the adsorbed water in the non-annealed CdSe_0.6_Te_0.4_. In the spectrum of the annealed sample, the O–H peak disappeared because the water was volatilized during annealing at 350 °C. In Figure 1f, the characteristic peaks at 2832, 2920, and 2959 cm^−1^ in the spectra of both CdSe_0.6_Te_0.4_ samples correspond to CH_2_–S bonding [40]. Peaks related to the asymmetric vibrational mode of the C–O bonds are observed at 1732 and 1638 cm^−1^ in the spectra of both the samples. These two peaks confirm the presence of L-cys on CdSe_0.6_Te_0.4_ [44]. The spectra of both the non-annealed and annealed CdSe_0.6_Te_0.4_ samples show a vibrational peak at 1452 cm^−1^, which is related to the symmetric stretching vibration of the C–N bonds [45,46]. The stronger peaks in the range of 1000–1374 cm^−1^ correspond to a Cd-Se-Te composite formed in both the non-annealed and annealed CdSe_0.6_Te_0.4_ samples. The FT-IR spectra confirm the formation of pure-phase CdSe_0.6_Te_0.4_ thin films. 

### 2.4. FE-SEM

FE-SEM analyses were used to study the surface morphology of the non-annealed CdSe_0.6_Te_0.4_ samples as well as the effect of the annealing on their surface morphology. Figure 2a–h presents the FE-SEM analyses of the non-annealed CdSe_0.6_Te_0.4_ samples and those annealed at different temperatures (300, 350, and 400 °C) with different magnifications. All the samples are fully covered with uniform thin films of various nanostructures with cauliflower-like, broccoli-like, and coral-like morphologies with various temperatures. Figure 2a,b presents the FE-SEM images of the non-annealed CdSe_0.6_Te_0.4_ samples at different magnifications. At higher magnifications, cauliflower-like nanostructures are clearly observed to fully cover the surface of the samples [32,33,34]. The FE-SEM images of the CdSe_0.6_Te_0.4_ samples annealed in air exhibit various surface morphologies, as shown in Figure 2c–h. The surface of the CdSe_0.6_Te_0.4_ sample annealed at 300 °C exhibits vertical nanorod-like nanostructures. Further, the growth of nanoleaves is clearly observed at the top of the nanorods, possibly because of the annealing treatment [47,48,49]. These results suggest that annealing is useful for developing various surface structures. 

When we increased the annealing temperature from 300 to 350 °C, the treated CdSe_0.6_Te_0.4_ sample exhibited a unique, porous morphology, as shown in Figure 2e,f. The images clearly show that the surface is highly porous and that one end of the tubes is open, which is useful for PEC applications. The image of the other end shows flower-bouquet-like morphology developed on the nanotubes. Similar morphology has been reported to strongly influence PEC performance because these types of morphology provide a larger surface area [33]. Further, when the annealing temperature increased to 400 °C, samples with tree-branch-like morphology were observed, as shown in Figure 2g,h. The surface morphological studies confirmed that the annealing temperature affected the porosities, morphology, and active surface areas of the CdSe_0.6_Te_0.4_ sample [34,50]. The schematic formation and effect of the annealing temperature on the CdSe_0.6_Te_0.4_ nanostructures are presented in Figure 1.

Figure 3a–h shows the EDS spectra and elemental mapping images of the non-annealed CdSe_0.6_Te_0.4_ and the air-annealed sample at 350 °C, respectively. The EDS analysis showed that Cd, Se, and Te elements are present in both the non-annealed and optimized CdSe_0.6_Te_0.4_ samples. The EDS analysis also revealed the presence of O in both the samples because all the samples were prepared in water; the XPS results supported the EDS analysis. Moreover, Pt was observed because a Pt coating was applied to the samples for the FE-SEM analysis [34,35,36,37]. 

### 2.5. Band Gap Energy Study

Figure 4 shows the optical band gap energy plots of the non-annealed CdSe_0.6_Te_0.4_ samples and the annealed ones at 300, 350, and 400 °C. The optical band gap energy was obtained using the following Equation (1) [35]:(1)$$\alpha =\frac{A{\left(h\nu -E_{g}\right)}^{n/2}}{h\nu }$$
where A and n are constants, “hν” is the photon energy, and E_g_ is the band gap energy. The direct and indirect band gap energies of semiconductor materials depend on the aforementioned values. The band gap energy results indicate that the non-annealed and annealed CdSe_0.6_Te_0.4_ samples are semiconductor materials with direct band gap energies. 

The E_g_ value of the non-annealed CdSe_0.6_Te_0.4_ samples was 1.70 eV; after annealing the CdSe_0.6_Te_0.4_ samples at 300 °C, the band gap energy slightly increased to 1.72 eV. Furthermore, by increasing the annealing temperature from 300 to 350 °C, the band gap energy decreased to 1.55 eV. The optical band gap energy value decreased with increasing air annealing temperature; this change is attributable to the changes in the nanostructures and thermal defects of the CdSe_0.6_Te_0.4_ thin films. However, when we studied the effect of the higher temperature on CdSe_0.6_Te_0.4_ thin films at 400 °C, the band gap energy increased up to 1.69 eV. The version of the band gap energy values demonstrated that redshift occurs in all the CdSe_0.6_Te_0.4_ thin films [21,51,52]. Muthukumarasamy et al. [53] prepared the CdSe_x_Te_1−x_ ternary composite on conducting glass substrates using the hot-wall deposition method with a varying composition of Se and Te precursors (*x* = 0.15, 0.4, 0.6, 0.7, 0.85). They reported that band gap energies increased in the range from 1.54 to 1.72 eV. Velumani et al. [54] reported that band gap energies increased between 1.52 and 1.62 eV, and Sathyamoorthy et al. [55] reported band gap energies in the ranges of 1.52 to 2.66 eV. These reported values of the band gap energies are in good agreement with the reported values in the present work.

### 2.6. Photoelectrochemical Study

Figure 5 shows the *J*-*V* measurement curves for the non-annealed and the annealed CdSe_0.6_Te_0.4_ photoelectrodes at different temperatures in the range from 300 to 400 °C, respectively. The photocurrent density of the non-annealed CdSe_0.6_Te_0.4_ photoelectrode was 1.91 mA cm^2^, and the photocurrent densities of the CdSe_0.6_Te_0.4_ photoelectrodes annealed at 300, 350 and 400 °C were 2.59, 2.96, and 2.15 mA cm^2^, respectively. After annealing in air, the CdSe_0.6_Te_0.4_ photoelectrode showed measurable changes in the photocurrent density and minor changes in the photovoltage, where the magnitude of the changes depends on the annealing temperature of the CdSe_0.6_Te_0.4_ photoelectrode. These results suggest that the annealing temperatures influence the photocurrent density of all the annealed CdSe_0.6_Te_0.4_ photoelectrodes. The *J*-*V* measurement presented that the CdSe_0.6_Te_0.4_ photoelectrode annealed at 350 °C exhibits the highest photocurrent density of 2.96 mA cm^2^, which is attributed to its highly porous and hollow-nanowire-like surface morphology. The fill factor (FF) was obtained by the following standard relation Equation (2) [48]:(2)$$\mathrm{FF}=\frac{I_{\max}\times V_{\max}}{V_{oc}\times I_{sc}}$$

Another significant feature of solar cells is their efficiency (η). The solar cell ƞ is directly related to the ratio of the output power to the incident light on the working photoelectrodes. The ƞ was obtained using the standard Equation (3) [31,33]:(3)$$\eta (\%)=\frac{V_{\mathrm{oc}}\times I_{\mathrm{sc}}}{P_{\mathrm{input}}}\times \mathrm{FF}\times 100$$

The values of I_max_, V_max_, I_sc_, V_oc_, FF, and η for the non-annealed and annealed CdSe_0.6_Te_0.4_ photoelectrodes are reported in Table 1. Table 1 presents the values of power ƞ of the non-annealed and annealed CdSe_0.6_Te_0.4_ photoelectrodes as 2.0, 3.1, 3.6, and 2.5 %, respectively. The resultant values of ƞ indicate that temperatures are positively affected on the ternary metal chalcogenides CdSe_0.6_Te_0.4_ photoelectrodes. The value of η increased with increasing annealing temperatures up to 350 °C and then decreased when the annealing temperature was further increased to 400 °C, which may be due to the band gap energy and porous nanostructures [35,37]. The lower band gap energy and highly porous surface morphology of photoelectrodes provided more electron–hole pairs and higher surface areas as compared to the other photoelectrodes for PEC application. The results of the η values indicate that the annealing temperature affects both the surface morphology and the solar cell performance. The CdSe_0.6_Te_0.4_ sample annealed at 350 °C exhibited the best solar cell performance among the investigated samples because of its highly porous, nanowire-like surface structure, which provides abundant active sites for the separation of charge carriers [30,31,32,33]. 

## 3. Materials and Methods 

### 3.1. Materials

The reagents used to prepare the CdTe_0.6_Se_0.4_ samples were 0.1 M cadmium sulfate monohydrate (CdSO_4_·H_2_O), 0.01 M selenium dioxide (SeO_2_), 0.02 M sodium telluride (Na_2_TeO_3_), and EDTA. All the chemicals were purchased from Sigma-Aldrich (Incheon, Korea).

### 3.2. Synthesis of CdTe_0.6_Se_0.4_ Thin Films 

The CdSe_0.6_Te_0.4_ was deposited on stainless steel (SS) and fluorine-doped tin oxide (FTO) glass by using the potentiostatic method. Precleaned SS and FTO were used as working electrodes while SCE and graphite served as reference and counter electrode, respectively. The dried substrates with an area of 1 cm^2^ were immersed in an electrolyte solution containing CdSO_4_·8H_2_O, SeO_2_, Na_2_TeO_3_, and EDTA to deposit the CdSe_0.6_Te_0.4_. After 30 min, the substrates were removed and cleaned using DDW. The non-annealed and annealed samples at 300, 350, and 400 °C are denoted as CST-000, CSTA-300, CSTA-350, and CSTA-400, respectively [33,37]. 

### 3.3. Characterization Techniques

A Philips PW-3710 X-ray powder diffractometer (λ = 1.54 Å, Cu-K_α_) was used to measure the structural properties of the samples in the range of 20–80°. FE-SEM was then conducted using a JEOL JSM-6360 to observe the surface morphologies of all samples. The optical properties of the composite samples deposited on the FTO substrates were studied using a Systronics model 119 spectrophotometer. The XPS analysis was performed using a Physical Electronics 5600 Multi-technique system equipped with a monochromatic Al K_α_ radiation source to analyze the surface compositions of the samples [36,37].

### 3.4. Electrode Preparation and PEC Measurements 

The PECs were fabricated in a two-electrode configuration comprising n-CdSe_0.6_Te_0.4_ composites as a photoanode and graphite as a counter electrode; 1 M polysulfide (NaOH:Na_2_S:S) was used as the redox electrolyte. The PEC cell was illuminated with a 100 W tungsten-filament lamp (I = 30 mW cm^−2^) to measure the short-circuit current (I_sc_) and open-circuit voltage (V_oc_).

## 4. Conclusions

Here, we systematically investigated the effect of different annealing temperatures on CdSe_0.6_Te_0.4_ composite samples for use in PEC cells. All the non-annealed and annealed samples were polycrystalline in nature with a hexagonal structure. The FE-SEM analysis showed that the surface morphologies can be altered by the annealing temperature. The surface morphologies such as cauliflower-like, broccoli-like, and coral-like structures were developed on the surface of the substrate. The optical band gap of the CdSe_0.6_Te_0.4_ samples was in the range of 1.70–1.55 eV, suggesting that the ternary CdSe_0.6_Te_0.4_ composite is suitable for use in PEC solar cells. The maximum η and FF of this PEC cell were 3.6 % and 0.53, respectively, as observed for the photoelctrode annealed at 350 °C. 

## Data Availability

The data presented in this study are available on request from the corresponding author. The data are not publicly available due to privacy. Data is contained within the article or Appendix A.

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
