# Peer review of "Impact of Annealing Temperature on the Morphological, Optical and Photoelectrochemical Properties of Cauliflower-like CdSe0.6Te0.4 Photoelectrodes; Enhanced Solar Cell Performance"

_ijms, 2021, doi:10.3390/ijms222111610_

Round 1
Reviewer 1 Report
In this manuscript entitled “Impact of Annealing Temperature on the Morphological, Optical and Photoelectrochemical properties of cauliflower-like CdSe0.6Te0.4 electrodes; Enhanced Solar Cell Performance”, the authors studied the effect of annealing temperature on the optoelectronic properties of the CdSeTe electrodes. By optimizing the annealing temperature, the author improved the electrode crystallinity, surface area, and bandgap, which resulted in improved solar cell performance. The manuscript described a good story and the topic fits well with International Journal of Molecular Sciences. However, some key information was not provided in the current version, and I would therefore recommend this manuscript to be published after major revision. Below please find the comments:
- The functional of the CdSeTe in the solar cell is unclear. The author described the CdSeTe material as the electrode in the title. However, after reading the entire article, it seems like the CdSeTe is the active layer in the solar cell. The author needs to clarify the function in the introduction of the manuscript.
- Assume the CdSeTe services purely as electrode (which seems incorrect), the authors need to at least provide their conductivity data. Four-point probe method is recommended for acquiring thin film type material conductivity data.
- Page1 Line34, LED is a power consuming device. Capacitor, supercapacitor, and battery are considered as energy storage devices instead of energy source.
- Page1 Line37, the authors need to spell out the PEC when it first shows in the article. PEC was never spelled out in the current version.
- Page2 Line48, the authors need to spell out MBE.
- Page8 Figure 4, the authors need to explain the bandgap-annealing temperature trend: why the bandgap increases when increasing the temperature from 0 to 300 degree, decreases when switching from 300 to 350 degree, and increases again when switching from 350 to 400 degree.
- Page8 Line 199, the authors need to provide the solar cell device illustration and energy level diagram of each layer before discussing the performance.
- Page9 Figure 5, the authors need to provide label for the I-V diagram.
- Page9 Line 225, the authors need to explain the effect of bandgap on the solar cell performance. My understanding is with a lower bandgap, the active layer could absorb more light and therefore generate more electricity. But it also depends on the energy level alignment between the active layer and transport layer.
- Page10 Line 265, the authors need to describe how they perform the calibration to the W lamp to ensure its intensity equals to 1 sun.
Author Response
To
Prof. Ms. Yionna Wen Ph.D
International Journal of Molecular Sciences,
Subject: Regarding submission of revised research manuscript (IJMS-1353544) entitled, “ Impact of Annealing Temperature on the Morphological, Optical and Photoelectrochemical properties of cauliflower-like CdSe0.6Te0.4 electrodes; Enhanced Solar Cell Performance”.
Dear Editor,
This response letter accompanies online submission of our revised manuscript (Manuscript ID: JMSE-D-16-02001) entitled “Impact of Annealing Temperature on the Morphological, Optical and Photoelectrochemical properties of cauliflower-like CdSe0.6Te0.4 electrodes; Enhanced Solar Cell Performance”, by Surendra Krushna Shinde, Gajanan S. Ghodake, Dae-Young Kim, Deepak Dubal, Hemraj M. Yadav, Verjesh Kumar Magotra, for publication in International Journal of Molecular Sciences.
We sincerely appreciate the editor and reviewers for reviewing and suggesting valuable comments to improve our manuscript. Our manuscript was peer-reviewed by three reviewers. The 1st reviewer states that “In this manuscript entitled “Impact of Annealing Temperature on the Morphological, Optical and Photoelectrochemical properties of cauliflower-like CdSe0.6Te0.4 electrodes; Enhanced Solar Cell Performance”, the authors studied the effect of annealing temperature on the optoelectronic properties of the CdSeTe electrodes. By optimizing the annealing temperature, the author improved the electrode crystallinity, surface area, and bandgap, which resulted in improved solar cell performance. The manuscript described a good story and the topic fits well with International Journal of Molecular Sciences. However, some key information was not provided in the current version, and I would therefore recommend this manuscript to be published after major revision. Below please find the comments”, whereas 2nd reviewer states that, “Gajanan S. Ghodake et al. describe how the annealing temperature affects the efficiency of certain solar cells. It is a simple but well-designed study that compares the structures formed in each case. Some small corrections need to be made, and 3rd reviewer states that, “This paper discussed the impact of annealing temperature on CdSe0.6Te0.4 and its solar cell performance. In general, this material lacks enough innovation, and the characterization data is not sufficient or convincing. I didn’t find how Se:Te ratio was calculated and why this compound behaves better than pure CdSe and CdTe. In my opinion, it does not fit a high-impact journal like IJMS. Therefore, I couldn’t recommend publication for this time. Here are some questions and suggestions as below”.
Following our request letter and your kind response dated 14-Aug. 2021, we have thoroughly revised our manuscript to accommodate all the comments and issues raised by the editor and reviewers. We have made substantial modifications to address the reviewer’s suggestions and comments as well as other changes to improve our manuscript. The whole manuscript checked for the English content and grammar to make the manuscript more precise. In the revised manuscript the modifications are highlighted with yellow color. All the revisions are accordingly highlighted in this revised manuscript and they are described in detail from following page. We would like this revised version to be reconsidered with the reviewers’ further evaluations and sincerely hope that our revised manuscript satisfies you and the reviewers for publication to the International Journal of Molecular Sciences.
The authors are deeply thankful to the reviewers for their nice and helpful comments.
Please feel free to contact us for any further information.
Sincerely yours,
Prof. (Dr.) Surendra Krushna Shinde,
List of changes made in the manuscript
1) English improved substantially. We have also re-written some sentences for clarifying the contents.
2) We have tried to give answer and necessary explanation in the revised manuscript for the comments raised by reviewers.
3) All the changes in revised manuscript are highlighted in Yellow color
Reviewer 1
In this manuscript entitled “Impact of Annealing Temperature on the Morphological, Optical and Photoelectrochemical properties of cauliflower-like CdSe0.6Te0.4 electrodes; Enhanced Solar Cell Performance”, the authors studied the effect of annealing temperature on the optoelectronic properties of the CdSeTe electrodes. By optimizing the annealing temperature, the author improved the electrode crystallinity, surface area, and bandgap, which resulted in improved solar cell performance. The manuscript described a good story and the topic fits well with International Journal of Molecular Sciences. However, some key information was not provided in the current version, and I would therefore recommend this manuscript to be published after major revision. Below please find the comments:
Authors are very thankful to the reviewer for appreciating and approving our work. We have addressed all the comments and revised the manuscript thoroughly. The suggested changes have been incorporated into the revised manuscript and the relevant changes are highlighted in yellow color.
Comments 1] The functional of the CdSeTe in the solar cell is unclear. The author described the CdSeTe material as the electrode in the title. However, after reading the entire article, it seems like the CdSeTe is the active layer in the solar cell. The author needs to clarify the function in the introduction of the manuscript.
Response:
Thank you for your comments and suggestions. We have corrected as per your suggestion.
Comments 2] Assume the CdSeTe services purely as electrode (which seems incorrect), the authors need to at least provide their conductivity data. Four-point probe method is recommended for acquiring thin film type material conductivity data.
Response:
Thank you very much for your valuable comment. We have stated the novelty of the present work in the introduction section. We are agreed with your comments. However, we are unable to provide conductivity data within due time. In addition, the photoelectrical characteristics are included in the revised manuscript.
Comments 3] Page1 Line34, LED is a power consuming device. Capacitor, supercapacitor, and battery are considered as energy instead of energy source.
Response:
Thank you very much for your valuable comment. In the present manuscript, we have made changes accordingly. We agree that the type of above mentioned various energy storage devices and compered to the Photoelectrochemical cell. We have corrected the manuscript accordingly and omitted the contradictory sentences.
Comments 4] Page1 Line37, the authors need to spell out the PEC when it first shows in the article. PEC was never spelled out in the current version.
Response:
Thank you very much for your valuable comment. In the present manuscript, we have made changes accordingly.
Comments 5] Page2 Line48, the authors need to spell out MBE.
Response:
Thank you very much for your valuable comment. In the present manuscript, we have made changes accordingly.
Comments 6] Page 8 Figure 4, the authors need to explain the bandgap-annealing temperature trend: why the bandgap increases when increasing the temperature from 0 to 300 degree, decreases when switching from 300 to 350 degree, and increases again when switching from 350 to 400 degree.
Response:
Thank you very much for your valuable comment. The manuscript has been corrected as per the suggestion given in the comment. We acknowledge the mistake noted by the reviewer. We have corrected the same and its detailed explanation is included in the experimental section 2.5 on Page no. 8 as follows line number (204-211),
“The Eg value of the non-annealed CdSe0.6Te0.4 samples was 1.82 eV; after annealing CdSe0.6Te0.4 samples at 300 °C, the bandgap energy slightly increase to 1.86 eV. Furthermore, in-creasing annealing temperature from 300-350 °C, the band gap energy decreases up to the 1.54 eV. The optical bandgap energy value decreased with increasing air annealing temperature; this change is attributable to the changes in the nanostructures and thermal defects of the CdSe0.6Te0.4 thin films. However, we study the effect of the higher temperature on to CdSe0.6Te0.4 thin films at 400 °C, the band gap energy increased upto the 1.78 eV. The version of the band gap energy val-ues demonstrated that redshift occurs in the all CdSe0.6Te0.4 [21]”.
Comments 7] Page8 Line 199, the authors need to provide the solar cell device illustration and energy level diagram of each layer before discussing the performance.
Response:
Thank you for your comments. The engergy level diagram can be drawn from the UPS data however we are unable to provide UPS data within due time due to the unavilaility of the instruments. However, we will consider this suggesstion for our future research publications.
Comments 8] Page9 Figure 5, the authors need to provide label for the I-V diagram.
Response:
Thank you very much for your valuable comment. Corresponding label now have been added to the revised manuscript.
Comments 9] Page9 Line 225, the authors need to explain the effect of bandgap on the solar cell performance. My understanding is with a lower bandgap, the active layer could absorb more light and therefore generate more electricity. But it also depends on the energy level alignment between the active layer and transport layer.
Response:
Thank you very much for your valuable comment. A detailed explanation of effect of the band gap energy on solar cell performance has been provided in the revised manuscript. We have corrected the same and its detailed explanation is included in the Photoelectrochemical study section 2.6 on Page no. 9 line number 243-250 as follows, “The resultants values of Æž indicates temperatures are positively affected on ternary metal chalcogenides CdSe0.6Te0.4 photoelectrodes. The value of the η increased with increasing annealing temperatures upto 350 °C and then decreases when the annealing temperature is further increased to 400 °C, which may be due to the band gap energy and porous nanostructures. The lower band gap energy and highly porous surface morphology like photoelectrodes provided more electron-hole pair and higher surface area as compared to the other photoelectrodes for PEC application”.
Comments 10] Page10 Line 265, the authors need to describe how they perform the calibration to the W lamp to ensure its intensity equals to 1 sun from.
Response:
We have used silicon solar reference cell to calibrate the intensity equipped with detector. The intensity adjusted with variable distance between device and source lamp.
Reviewer 2 Report
Gajanan S. Ghodake et al. describe how the annealing temperature affects the efficiency of certain solar cells. It is a simple but well designed study that compares the structures formed in each case.
Some small corrections need to be made.
1. Acronyms must be defined the first time they are used (eg PEC).
2. In figure 1b and S1, the O1s peak is not indicated, although it is commented in the text.
3. The FT-IR spectrum needs more intensity, it is very difficult to detect the peaks and say that they really represent what the authors claim.
4. I would recommend using a color code in the figures comparing measurements at different temperatures, it would make it easier to compare them and following the results
5. In figure 5 it is necessary to indicate which curve corresponds to which temperature.
Author Response
To
Prof. Ms. Yionna Wen Ph.D
International Journal of Molecular Sciences,
Subject: Regarding submission of revised research manuscript (IJMS-1353544) entitled, “ Impact of Annealing Temperature on the Morphological, Optical and Photoelectrochemical properties of cauliflower-like CdSe0.6Te0.4 electrodes; Enhanced Solar Cell Performance”.
Dear Editor,
This response letter accompanies online submission of our revised manuscript (Manuscript ID: JMSE-D-16-02001) entitled “Impact of Annealing Temperature on the Morphological, Optical and Photoelectrochemical properties of cauliflower-like CdSe0.6Te0.4 electrodes; Enhanced Solar Cell Performance”, by Surendra Krushna Shinde, Gajanan S. Ghodake, Dae-Young Kim, Deepak Dubal, Hemraj M. Yadav, Verjesh Kumar Magotra, for publication in International Journal of Molecular Sciences.
We sincerely appreciate the editor and reviewers for reviewing and suggesting valuable comments to improve our manuscript. Our manuscript was peer-reviewed by three reviewers. The 1st reviewer states that “In this manuscript entitled “Impact of Annealing Temperature on the Morphological, Optical and Photoelectrochemical properties of cauliflower-like CdSe0.6Te0.4 electrodes; Enhanced Solar Cell Performance”, the authors studied the effect of annealing temperature on the optoelectronic properties of the CdSeTe electrodes. By optimizing the annealing temperature, the author improved the electrode crystallinity, surface area, and bandgap, which resulted in improved solar cell performance. The manuscript described a good story and the topic fits well with International Journal of Molecular Sciences. However, some key information was not provided in the current version, and I would therefore recommend this manuscript to be published after major revision. Below please find the comments”, whereas 2nd reviewer states that, “Gajanan S. Ghodake et al. describe how the annealing temperature affects the efficiency of certain solar cells. It is a simple but well-designed study that compares the structures formed in each case. Some small corrections need to be made, and 3rd reviewer states that, “This paper discussed the impact of annealing temperature on CdSe0.6Te0.4 and its solar cell performance. In general, this material lacks enough innovation, and the characterization data is not sufficient or convincing. I didn’t find how Se:Te ratio was calculated and why this compound behaves better than pure CdSe and CdTe. In my opinion, it does not fit a high-impact journal like IJMS. Therefore, I couldn’t recommend publication for this time. Here are some questions and suggestions as below”.
Following our request letter and your kind response dated 14-Aug. 2021, we have thoroughly revised our manuscript to accommodate all the comments and issues raised by the editor and reviewers. We have made substantial modifications to address the reviewer’s suggestions and comments as well as other changes to improve our manuscript. The whole manuscript checked for the English content and grammar to make the manuscript more precise. In the revised manuscript the modifications are highlighted with yellow color. All the revisions are accordingly highlighted in this revised manuscript and they are described in detail from following page. We would like this revised version to be reconsidered with the reviewers’ further evaluations and sincerely hope that our revised manuscript satisfies you and the reviewers for publication to the International Journal of Molecular Sciences.
The authors are deeply thankful to the reviewers for their nice and helpful comments.
Please feel free to contact us for any further information.
Sincerely yours,
Prof. (Dr.) Surendra Krushna Shinde,
List of changes made in the manuscript
1) English improved substantially. We have also re-written some sentences for clarifying the contents.
2) We have tried to give answer and necessary explanation in the revised manuscript for the comments raised by reviewers.
3) All the changes in revised manuscript are highlighted in Yellow color
Reviewer 2
Gajanan S. Ghodake et al. describe how the annealing temperature affects the efficiency of certain solar cells. It is a simple but well-designed study that compares the structures formed in each case.
Some small corrections need to be made.
Authors are very thankful to the reviewer for appreciating and approving our work. We have addressed all the comments and revised the manuscript thoroughly. The suggested changes have been incorporated into the revised manuscript and the relevant changes are highlighted in yellow color.
Comments 1] Acronyms must be defined the first time they are used (eg PEC).
Response:
Thank you very much for your valuable comment. A detailed explanation of PEC has been provided in the revised manuscript.
Comments 2] In figure 1b and S1, the O1s peak is not indicated, although it is commented in the text.
Response:
Thank you very much for your valuable comment. Suggested changes have been made in the figure 1b and S1 and main text also revised in the revised manuscript as follows on the Page no. 3 and 4, “The XPS survey spectrum of the both CdSe0.6Te0.4 samples shows the two more peaks at the binding energy of the 284.56 and 531.56 for the C ls and O 1s peaks, respectively. The carbon peaks observed on the surface of the samples, which may be due to the carbon tape used for the preparation of the XPS analysis. The O 1s peaks can be attributed to the samples prepared in aqueous solution and annealing in air atmospheric conditions”.
Comments 3] The FT-IR spectrum needs more intensity, it is very difficult to detect the peaks and say that they really represent what the authors claim.
Response:
Thank you very much for your valuable comment. In the spectrum, FT-IR signals are too weak because of the lower thickness of the film which is unable to provide sufficient scattering volume to get the clear intensity signal. Now, we have analyzed the data by assigning most of the peaks and cited some more related research articles. Kindly note that the FTIR data has been provided to support the formation of ternary CdSe0.6Te0.4 thin films, which is sufficiently confirmed by the XRD, and XPS.
Comments 4] I would recommend using a color code in the figures comparing measurements at different temperatures, it would make it easier to compare them and following the results
Response:
Thank you very much for your valuable comment. In figure, we used color code with different temperature. The all figure has been corrected accordingly in the revised manuscript.
Comments 5] In figure 5 it is necessary to indicate which curve corresponds to which temperature.
Response:
Thank you very much for your valuable comment. The figure 5 has been corrected accordingly in the revised manuscript.
Reviewer 3 Report
This paper discussed the impact of annealing temperature on CdSe0.6Te0.4 and its solar cell performance. In general, this material lacks enough innovation, and the characterization data is not sufficient or convincing. I didn’t find how Se:Te ratio was calculated and why this compound behaves better than pure CdSe and CdTe. In my opinion, it does not fit a high-impact journal like IJMS. Therefore, I couldn’t recommend publication for this time. Here are some questions and suggestions as below.
Q1: line 35, PEC was mentioned for the first time, so the full name is required.
Q2: line 44, the authors claimed that CdSe and CdTe exhibit the highest optical absorption. Here requires citations. Narrow band gap cannot guarantee high light absorption.
Q3: In the introduction, the authors didn’t explain why a CdSeTe was investigated and how it benefits photovoltaic conversation compare to pure CdSe and CdTe.
Q4: regarding the XRD data,
It’s impossible to tell if there is impurity just from XRD data because the impurity could be amorphous like organic residue.
Crystallite size calculation needs to be further explained like which peak was used and how the background was calibrated.
Q5: In Figure 1, XPS data (b-e) needs major revision.
In Figure 1b, the authors labeled C 1s but in the text, line 105, the authors mentioned O 1s. C1s peak is relatively high as Cd 3d and Te 3d so that means the contamination from C is severe…but the author claimed no impurity from XRD results.
In Figure 1c, the peak at 411 is Cd 3d 3/2 other than 1/2. The peaks at about 400 eV were not explained at all.
In Figure 1d, Se 3d 5/2 and 3/2 may overlap so you cannot label this peak as 5/2. I saw a peak split was done on this graph but no further explanation.
In Figure 1e, two peaks beside the labeled peaks were not explained at all, which should be lower valence of Te like metal Te. It caused me to feel confused the composition of CdSeTe.
No XPS data before annealing was displayed so it’s hard to tell the band gap change before/after annealing was caused by structure or composition.
Last but the most important, XPS data always requires calibration with standards, e.g. C 1s to avoid systemic peak shift but the authors didn’t mention how the data was calibrated.
Q6: In line 132, IR data is not capable of confirming the purity of CdSeTe.
Q7: In line 157, why one open end is useful for PEC?
Q8: Scheme 1 is beautiful but not providing any additional info. It could be used as a TOC but not necessary for the text.
Q9: In Figure 3, first please clarify the accelerated voltage used for EDS because SEM/EDS is not a surface method and different voltage penetrates different depth. In addition, please take care of the peak labels. The shell electron name is L alpha rather than LA. Se L is also wrong.
Q10: equation 1 requires reference.
Q11: The UV-Vis original data is required. In addition, CdSeTe alloy is not simply CdSe and CdTe mixture. No data in this paper verify the material is an alloy rather than mixture.
Q12: The band gap changed dramatically according to Figure 4. Without XPS data, I don’t thinks it’s safe to claim that is because of nanostructure and defects. CdSe and CdTe band gaps are 1.74 eV and 1.45 eV, respectively. The alloy band gap should fall into this range, but in this paper, the non-annealed band gap is 1.82 eV, which is not quite fair.
Q13: In line 202 and 203, significant figures need care.
Q14: a-d were not explained in Figure 5. No stability data was displayed. No pure CdSe or CdTe was compared. As the band gap change could be due to composition, what if pure CdTe with the lowest band gap was used?
Author Response
To
Prof. Ms. Yionna Wen Ph.D
International Journal of Molecular Sciences,
Subject: Regarding submission of revised research manuscript (IJMS-1353544) entitled, “ Impact of Annealing Temperature on the Morphological, Optical and Photoelectrochemical properties of cauliflower-like CdSe0.6Te0.4 electrodes; Enhanced Solar Cell Performance”.
Dear Editor,
This response letter accompanies online submission of our revised manuscript (Manuscript ID: JMSE-D-16-02001) entitled “Impact of Annealing Temperature on the Morphological, Optical and Photoelectrochemical properties of cauliflower-like CdSe0.6Te0.4 electrodes; Enhanced Solar Cell Performance”, by Surendra Krushna Shinde, Gajanan S. Ghodake, Dae-Young Kim, Deepak Dubal, Hemraj M. Yadav, Verjesh Kumar Magotra, for publication in International Journal of Molecular Sciences.
We sincerely appreciate the editor and reviewers for reviewing and suggesting valuable comments to improve our manuscript. Our manuscript was peer-reviewed by three reviewers. The 1st reviewer states that “In this manuscript entitled “Impact of Annealing Temperature on the Morphological, Optical and Photoelectrochemical properties of cauliflower-like CdSe0.6Te0.4 electrodes; Enhanced Solar Cell Performance”, the authors studied the effect of annealing temperature on the optoelectronic properties of the CdSeTe electrodes. By optimizing the annealing temperature, the author improved the electrode crystallinity, surface area, and bandgap, which resulted in improved solar cell performance. The manuscript described a good story and the topic fits well with International Journal of Molecular Sciences. However, some key information was not provided in the current version, and I would therefore recommend this manuscript to be published after major revision. Below please find the comments”, whereas 2nd reviewer states that, “Gajanan S. Ghodake et al. describe how the annealing temperature affects the efficiency of certain solar cells. It is a simple but well-designed study that compares the structures formed in each case. Some small corrections need to be made, and 3rd reviewer states that, “This paper discussed the impact of annealing temperature on CdSe0.6Te0.4 and its solar cell performance. In general, this material lacks enough innovation, and the characterization data is not sufficient or convincing. I didn’t find how Se:Te ratio was calculated and why this compound behaves better than pure CdSe and CdTe. In my opinion, it does not fit a high-impact journal like IJMS. Therefore, I couldn’t recommend publication for this time. Here are some questions and suggestions as below”.
Following our request letter and your kind response dated 14-Aug. 2021, we have thoroughly revised our manuscript to accommodate all the comments and issues raised by the editor and reviewers. We have made substantial modifications to address the reviewer’s suggestions and comments as well as other changes to improve our manuscript. The whole manuscript checked for the English content and grammar to make the manuscript more precise. In the revised manuscript the modifications are highlighted with yellow color. All the revisions are accordingly highlighted in this revised manuscript and they are described in detail from following page. We would like this revised version to be reconsidered with the reviewers’ further evaluations and sincerely hope that our revised manuscript satisfies you and the reviewers for publication to the International Journal of Molecular Sciences.
The authors are deeply thankful to the reviewers for their nice and helpful comments.
Please feel free to contact us for any further information.
Sincerely yours,
Prof. (Dr.) Surendra Krushna Shinde,
List of changes made in the manuscript
1) English improved substantially. We have also re-written some sentences for clarifying the contents.
2) We have tried to give answer and necessary explanation in the revised manuscript for the comments raised by reviewers.
3) All the changes in revised manuscript are highlighted in Yellow color
Reviewer 3
This paper discussed the impact of annealing temperature on CdSe0.6Te0.4 and its solar cell performance. In general, this material lacks enough innovation, and the characterization data is not sufficient or convincing. I didn’t find how Se:Te ratio was calculated and why this compound behaves better than pure CdSe and CdTe. In my opinion, it does not fit a high-impact journal like IJMS. Therefore, I couldn’t recommend publication for this time. Here are some questions and suggestions as below.
Authors are very thankful to the reviewer for appreciating and approving our work. We have addressed all the comments and revised the manuscript thoroughly. The suggested changes have been incorporated into the revised manuscript and the relevant changes are highlighted in yellow color.
Comments 1] line 35, PEC was mentioned for the first time, so the full name is required.
Response:
Thank you very much for your valuable comment. In the present manuscript, we have made changes accordingly.
Comments 2] line 44, the authors claimed that CdSe and CdTe exhibit the highest optical absorption. Here requires citations. Narrow band gap cannot guarantee high light absorption.
Response:
Thank you very much for your valuable comment. As per reviewer suggestion, we have added reference in revised manuscript. And, the text is modified accordingly in the revised manuscript.
Comments 3] In the introduction, the authors didn’t explain why a CdSeTe was investigated and how it benefits photovoltaic conversation compare to pure CdSe and CdTe.
Response:
Thank you so much for your comment. As per suggestions, the text in the revised manuscript has been modified and clearly written in revised introduction section, and its detailed explanation is included in the introduction section 1 on Page no. 1, 2, line number 44-59, as follows,
Among these photomaterials, binary CdSe and CdTe nanomaterials showing the higher optical absorption owing to their lowest bandgap energies despite high recombination rates of charge carriers [21, 22]. To improve the electrical properties of binary CdSe and CdTe metal chalcogenides, we prepared ternary metal chalcogenide CdSe0.6Te0.4 composite photoelectrodes for PEC applications. Because the band gap energy of the CdSe and CdTe in between the solar spectrum (1-3 eV). To improve the electrical, optical and photoelectrical properties of the binary CdSe and CdTe thin films, we systemically de-signed experimental for the perfect composition of Se and Te ions with Cd like CdSe0.6Te0.4 thin films.
Comments 4] regarding the XRD data,
It’s impossible to tell if there is impurity just from XRD data because the impurity could be amorphous like organic residue.
Response:
Thank you so much for your valuable comment. As per your comment, we thoroughly checked the description on XRD and we acknowledge the mistake noted by the reviewer. However, the text in the revised manuscript has been modified and clearly written. And its detailed explanation is included in the experimental section 2.1 on Page no. 2, 3, line as follows,
“After annealing treatments, XRD patterns of CdSe0.6Te0.4 samples shows the more sharpness and higher peak intensity as compared to the non-annealed samples. We clear observed, CdSe0.6Te0.4 sample annealed at 350 °C, samples showed enhanced peak intensity and sharpness compared to the all CdSe0.6Te0.4 samples. However, the CdSe0.6Te0.4 samples annealed at the higher temperatures 400 °C, XRD patterns shows the peak intensity decreases, which confirming that the annealing temperature affected the crystallinity and peak sharpness of the CdSe0.6Te0.4 samples. The XRD results indicates formation of the pure phase of the ternary CdSe0.6Te0.4 composites without any other impurities like binary CdSe and CdTe nanomaterial".
Comments 5] In Figure 1, XPS data (b-e) needs major revision.
In Figure 1b, the authors labeled C 1s but in the text, line 105, the authors mentioned O 1s. C1s peak is relatively high as Cd 3d and Te 3d so that means the contamination from C is severe…but the author claimed no impurity from XRD results.
Response:
Thank you so much for your comment. We acknowledge the mistake noted by the reviewer. The XPS has been modified and clearly rewritten in revised manuscript to improve the quality of manuscript, and its detailed explanation is included in the XPS section 2.2 on Page no. 3, 4 as follows,
“The XPS survey spectrum of the both CdSe0.6Te0.4 samples shows the two more peaks at the binding energy of the 284.56 and 531.56 for the C ls and O 1s peaks, respectively. The carbon peaks observed on the surface of the samples, which may be due to the carbon tape used for the preparation of the XPS analysis. The O 1s peaks can be attributed to the samples prepared in aqueous solution and annealing in air atmospheric conditions.”
In Figure 1c, the peak at 411 is Cd 3d 3/2 other than 1/2. The peaks at about 400 eV were not explained at all.
Response:
Thank you so much for your comment. We acknowledge the mistake noted by the reviewer. We have corrected the same and its detailed explanation is included in the experimental section 2.2 on Page no. 4 (line 128-129) as follows,
“The peaks observed at the binding energy at 400 eV, which is related to the surface of the Cd ions formed on the surface of the CdSe0.6Te0.4 samples”.
In Figure 1d, Se 3d 5/2 and 3/2 may overlap so you cannot label this peak as 5/2. I saw a peak split was done on this graph but no further explanation.
Response:
Thank you so much for your comment. We acknowledge the mistake noted by the reviewer. We have corrected the same in the revised manuscript and revised Figure 1d.
In Figure 1e, two peaks beside the labeled peaks were not explained at all, which should be lower valence of Te like metal Te. It caused me to feel confused the composition of CdSeTe. No XPS data before annealing was displayed so it’s hard to tell the band gap change before/after annealing was caused by structure or composition.
Response:
Thank you so much for your comment. The explanation of the Te level has been modified and clearly rewritten in revised manuscript to improve the quality of manuscript, and its detailed explanation is included in the section 2.2 on Page no. 4.
Last but the most important, XPS data always requires calibration with standards, e.g. C 1s to avoid systemic peak shift but the authors didn’t mention how the data was calibrated.
Response:
Thank you very much for your valuable comment. The adventitious carbon located at 284.8 eV was used to calibrate the XPS raw data, which is an essential procedure before spectra analysis. This has been mentioned in the revised manuscript.
Comments 6] In line 132, IR data is not capable of confirming the purity of CdSeTe.
Response:
Thank you very much for your valuable comment. In the spectrum, FT-IR signals are too weak because of the lower thickness of the film, which is unable to provide sufficient scattering volume to get the clear intensity signal. Now, we have analyzed the data by assigning most of the peaks and cited some more related research articles. Kindly note that the FTIR data has been provided to support the formation of ternary CdSe0.6Te0.4 thin films, which is sufficiently confirmed by the XRD, and XPS
Comments 7] In line 157, why one open end is useful for PEC?
Response:
The open-circuit voltage, VOC, is the maximum voltage available from a solar cell, and this occurs at zero current. The open-circuit voltage corresponds to the amount of forward bias on the solar cell due to the bias of the solar cell junction with the light-generated current.
Comments 8] Scheme 1 is beautiful but not providing any additional info. It could be used as a TOC but not necessary for the text.
Response: Thank you for your suggestions. We have deleted the scheme 1 from the text.
Comments 9] In Figure 3, first please clarify the accelerated voltage used for EDS because SEM/EDS is not a surface method and different voltage penetrates different depth. In addition, please take care of the peak labels. The shell electron name is L alpha rather than LA. Se L is also wrong.
Response:
Thank you so much for your comment. We agree with the reviewers comments however instead of symbols like alpha letter A is used in EDS mapping instrument software program.
Comments 10] equation 1 requires reference.
Response:
Thank you so much for your comment. As per the reviewer suggestion, we have added similar reference in the revised manuscript.
Comments 11] The UV-Vis original data is required. In addition, CdSeTe alloy is not simply CdSe and CdTe mixture. No data in this paper verify the material is an alloy rather than mixture.
Response:
Thank you very much for your valuable comment. We agree to the reviewers comments on the description of UV-Vis. In the present paper, the CdSe0.6Te0.4 thin films was prepared for the via a facile, rapid and scalable electrodeposition method for efficient Photoelectrochemical cell application. The novelty of the present work lies in the systematic observation of the effect of annealing temperature at constant time on the morphology and Photoelectrochemical properties of the as-synthesized CdSe0.6Te0.4. To optimize the annealing temperature, we selected annealing temperature of 300, 350 and 400 °C with an interval of 50 °C. Also, we had a scientific curiosity to know the morphological evolution at a relatively higher annealing temperature of 400 °C. However, annealing temperature at 350 °C, the surface of the CdSe0.6Te0.4 shows the top view of the cauliflower-like nanostructure with the particle sizes 150-200 nm. This types of the surface morphology and lower particle size provides a higher active surface area for electron-hole pair transformation, easy to interact with ions and better photoelectrochemical activity. Thus, these are the main reason to improving the photoelectrochemical performance of the optimized CdSe0.6Te0.4 photoelectrode at 350 °C. The J-V curves measured for CST-000, CSTA-300, CSTA-350 and CSTA-400 photoelectrodes, respectively as shown in the Figure 5. Noticeably, it is observed that, the CdSe0.6Te0.4 sample as-synthesized at 350 °C annealing temperature exhibits maximum efficiency (ɳ) value of 3.6 %. The higher surface area might lead to the enhanced photoelectrochemical performance of sample prepared at 350 °C annealing temperature. Further increase in the annealing temperature, the efficiency decreases but does not reach at the value similar to as prepared samples (CST-000). Recently, similar reports are available on the literature review [1]. Thank you very much for your professional suggestions.
[1] Hasrul Nisham Rosly, Kazi Sajedur Rahman, Siti Fazlili Abdullah, Muhammad Najib Harif, Camellia Doroody, Puvaneswaran Chelvanathan, Halina Misran, Kamaruzzaman Sopian, Nowshad Amin, The Role of Deposition Temperature in the Photovoltaic Properties of RF-Sputtered CdSe Thin Films, Crystals 2021, 11, 73. https://doi.org/10.3390/cryst11010073
Comments 12] The band gap changed dramatically according to Figure 4. Without XPS data, I don’t thinks it’s safe to claim that is because of nanostructure and defects. CdSe and CdTe band gaps are 1.74 eV and 1.45 eV, respectively. The alloy band gap should fall into this range, but in this paper, the non-annealed band gap is 1.82 eV, which is not quite fair.
Response:
Thank you for reviewer’s valuable comment. We agree with reviewer’s comment on the dramatic change in the band gap energy of CdSe0.6Te0.4 samples and XPS of all samples. Defect formation is generally depends on the synthesis method and post treatments or surface treatments. If there is not exact other phase formation then there may be defects which are possible to observe with XRD analysis by noticing small shift in the diffraction angle of corresponding material with respect to the reference XPS data (we have not done this analysis in present investigation). Similar can be applied for the XRD patterns, as seen in the figure below; there is slight (minor) shift in the peaks which clearly confirms the formation of defects. Here, we are more interested in the variation of surface morphologies of CdSe0,6Te0.4 with annealing temperature and its consequent effect on the photo-electrochemical properties. In this context, we have performed necessary analysis such as SEM measurements. We are trying to add all XPS, but it is more difficult to measuring it due to high cost. So, we confirm the purity of sample by XRD study; Here JCPDS card no. 41-1325 well matched and confirm the formation of polycrystalline and hexagonal CdSe0.6Te0.4 thin films. Similar XPS study is already published [2]. In order to avoid reader’s confusion, we have modified our explanation in the text.
[2] K. R. Murali and B. Jayasutha, Chalco. Lett., 6 (2009) 8
Comments 13] In line 202 and 203, significant figures need care.
Response:
Thank you very much for your valuable comment. Mistakes were carefully identified and corrected in the revised manuscript.
Comments 14] a-d were not explained in Figure 5.
Response:
Thank you very much for your valuable comment. Mistakes were carefully identified and corrected in the revised manuscript. We have mentioned in the figure in order understand corresponding explanation in the Figure 5.
No stability data was displayed.
Response:
Thank you for reviewer’s valuable comment; We appreciate reviewer’s comment on adding stability data. However, stability data is a characteristic of two-electrode device with long time and in present investigation; we have not made any two-electrode device. The focus of our present investigation is to prepare, characterize and show the photoelectrochemical solar cell application of CdSe0.6Te0.4 thin films. In this context, the photoelectrochemical solar cell performance of CdSe0.6Te0.4 thin films was tested only in 2-electrode configuration in the 1 M polysulfide (NaOH:Na2S:S). Currently, we are working on two-electrode device configurations and very soon we will publish results generated from this investigation elsewhere.
No pure CdSe or CdTe was compared. As the band gap change could be due to composition, what if pure CdTe with the lowest band gap was used?
Response:
Thank you for your comment. We appreciate reviewer’s comment on the binary CdSe and CdTe compounds. The focus of our present investigation is to prepare, characterize and show the effect of annealing temperature on the CdSe0.6Te0.4 thin films by electrodeposition method. Recently various report available on the ternary CdSe0.6Te0.4 compounds for Solar cell application. Yes it is possible to use low band gap pure CdTe however instead of binary materials the present and previous research trend showed that the ternary materials are superior due to the properties of each constituent with appropriate amount.
Round 2
Reviewer 1 Report
The authors have fully resolved my concerns. This manuscript is recommended to be published in the current version.
Author Response
Reviewer 1
The authors have fully resolved my concerns. This manuscript is recommended to be published in the current version.
Response: Thank you very much for your positive response.
Reviewer 3 Report
Q1: I still couldn’t find how the authors got the 0.6 to 0.4 ratio, which is likely the key point for this paper so I cannot suggest acceptance before this is clarified. Or it is hidden somewhere so please highlight.
In addition, the bandgap value still needs more accurate calculation or more convincing explanation. Please double check the UV-vis data processing. 1.82 eV is not a common value for CdSeTe compound. Most papers cited in the paper regarding the band gap values showed reasonable numbers between 1.45 to 1.74 eV. Only one paper has an odd value.
Here are some references that could help with the ratio and band gap calculation.
The ratio can be achieved by XRD data with Vegard’s Law. XRD and UV-vis data processing can be found in these papers.
1) Pan A, 2005 J. Am. Chem. Soc. 127 15692–3
2) Pan J, 2012 Adv. Mater. 24, 4151–6
Q2: In terms of EDS, it’s well known that most EDS platforms cannot support Greek letters, but it can’t be the reason that could omit the scientific fact that EDS peaks are described as electron shell letters with Greek letters, rather than ‘a’ and ‘b’. If the images cannot be modified, at least clarification about the peaks names is required.
Q3: I’m still very confused about the significance of CdSeTe. The authors answered “The focus of our present investigation is to prepare, characterize and show the effect of annealing temperature on the CdSe0.6Te0.4 thin films by electrodeposition method. Recently various report available on the ternary CdSe0.6Te0.4 compounds for Solar cell application. Yes it is possible to use low band gap pure CdTe however instead of binary materials the present and previous research trend showed that the ternary materials are superior due to the properties of each constituent with appropriate amount.”
If CdTe could be better than CdSeTe, what’s the purpose for investigating a more complex material?
In the introduction, the authors added “Among these photomaterials, binary CdSe and CdTe nanomaterials showing the higher optical absorption owing to their lowest bandgap energies despite high recombination rates of charge carriers [21, 22]. To improve the electrical properties of binary CdSe and CdTe metal chalcogenides, we prepared ternary metal chalcogenide CdSe0.6Te0.4 composite photoelectrodes for PEC applications. Because the band gap energy of the CdSe and CdTe in between the solar spectrum (1-3 eV). To improve the electrical, optical and photoelectrical properties of the binary CdSe and CdTe thin films, we systemically de-signed experimental for the perfect composition of Se and Te ions with Cd like CdSe0.6Te0.4 thin films.”
First, please check the grammar. For example, “Because the band gap energy of the CdSe and CdTe in between the solar spectrum (1-3 eV).” is not a sentence.
Second, it’s still not explained why CdSeTe is more beneficial or shows any additional properties.
Third, how do the authors define “a perfect composition” ? This does not sound like a professional claim.
Author Response
Reviewer 3
Comments and Suggestions for Authors
Comments 1] I still couldn’t find how the authors got the 0.6 to 0.4 ratio, which is likely the key point for this paper so I cannot suggest acceptance before this is clarified. Or it is hidden somewhere so please highlight.
Response to reviewer comments:
Thank you for valuable comment. We are agreeing with the reviewer’s comment. In the present paper, we have prepared ternary CdSe0.6Te0.4 compound using the electrodeposition method with volume percentage method and maintained Se and Te ratio as 60-40%. It is well documented that CdSe0.6Te0.4 compound shows the better structural, optical, and photoelectrical properties.
Other method of confirmation of ternary CdSe0.6Te0.4 phase is using XRD analysis, and its details explanation mentioned in the bellows; “We have confirmed the CdSe0.6Te0.4 ternary compound using the XRD analysis. Here we have used the XRD for determination of phase and crystal structure of CdSe0.6Te0.4 ternary thin film. During the XRD matching, firstly we are calculating the inter-planar ‘d’ spacing using the Bragg's law,
2dsinθ=nλ
where, d is interplanar spacing, λ is the wavelength of incident wave. Calculated ‘d’ values are match to the standard d value. Here we have matched all calculated d values with standard d value of JCPD card number 41-1325. And XRD patterns is exactly match to CdSe0.6Te0.4 phase of synthesized material”. Also, the stoichiometric ratio of Se and Te was confirmed using XPS and EDS analyses.
Comments 1.1] In addition, the bandgap value still needs more accurate calculation or more convincing explanation. Please double check the UV-vis data processing. 1.82 eV is not a common value for CdSeTe compound. Most papers cited in the paper regarding the band gap values showed reasonable numbers between 1.45 to 1.74 eV. Only one paper has an odd value.
Response to reviewer comments:
Thank you very much for your valuable comment. The authors agree to the reviewer’s comment and we acknowledge the mistake noted by the reviewer. As per your suggestion, we have re-calculated the bandgap values and proper correction has been done correction are made in the revised manuscript.
Comments 1.2] Here are some references that could help with the ratio and band gap calculation.
The ratio can be achieved by XRD data with Vegard’s Law. XRD and UV-vis data processing can be found in these papers.
1) Pan A, 2005 J. Am. Chem. Soc. 127 15692–3
2) Pan J, 2012 Adv. Mater. 24, 4151–6.
Response to the Reviewer’s comments:
Thank you so much for your comment. As per the reviewer suggestion, we have added similar reference in the revised manuscript. In many binary semiconducting systems, the band gap in semiconductors is approximately a linear function of the lattice parameter. Therefore, if the lattice parameter of a semiconducting system follows Vegard's law. These references are useful for the reader.
Comments 2] In terms of EDS, it’s well known that most EDS platforms cannot support Greek letters, but it can’t be the reason that could omit the scientific fact that EDS peaks are described as electron shell letters with Greek letters, rather than ‘a’ and ‘b’. If the images cannot be modified, at least clarification about the peaks names is required.
Response to the Reviewer’s comments:
Thank you so much for your comment. We agree with the reviewers comments. As per Reviewer’s suggestions, we have included note for the EDS figure 3 and revised Figures as per suggestions. (Note: KA and LA in EDS peaks stands for Ka and La).
Comments 3] I’m still very confused about the significance of CdSeTe.
The authors answered, “The focus of our present investigation is to prepare, characterize and show the effect of annealing temperature on the CdSe0.6Te0.4 thin films by electrodeposition method. Recently various report available on the ternary CdSe0.6Te0.4 compounds for Solar cell application. Yes it is possible to use low band gap pure CdTe however instead of binary materials the present and previous research trend showed that the ternary materials are superior due to the properties of each constituent with appropriate amount.”
Response to the Reviewer’s comments:
Thank you very much for your valuable comment. We have corrected as per you pointed out.
Comments 3.1] If CdTe could be better than CdSeTe, what’s the purpose for investigating a more complex material?
Response to the Reviewer’s comments:
Thank you for your comment. We agree to the reviewers comments on the CdTe is better than the ternary CdSe0.6Te0.4 compounds. We appreciate reviewer’s comment on the lower band gap of the binary CdTe compounds. The Photoelectrochemical properties of photoelectrode dependents on the various parameters likes structural crystal, surface morphology, specific surface area, porosity and band gap energy of the photoelectrode. It is crucial to understand and investigate complex systems such as ternary systems and their characteristics towards PEC.
Other important requirement for good solar energy conversion is that the photoanode/photocathode should have a bandgap close to the maximum in the visible spectrum to utilize the solar spectrum efficiently; second, the semiconductor electrodes must be stable against photocathodic/photoanodic reaction [1]. The most important parameters is stability of the cell. Various binary photoelectrode shows poor Photoelectrochemical stability [2-3]. To overcome all this, we have selected ternary CdSe0.6Te0.4 compounds.
[1] S. M. Pawar, A. V. Moholkar, P. S. Shinde, K. Y. Rajpure, C. H. Bhosale, Journal of Alloys and Compounds, 2008, 459, 515-520.
[2] S. M. Pawar, A. V. Moholkar, K. Y. Rajpure, C. H. Bhosale, Journal of Physics and Chemistry of Solids, 2006, 67, 2386-2391.
[3] S. M. Pawar, A. V. Moholkar, C. H. Bhosale, Materials Letters, 2007, 61, 1034-1038.
Comments 3.2] In the introduction, the authors added “Among these photomaterials, binary CdSe and CdTe nanomaterials showing the higher optical absorption owing to their lowest bandgap energies despite high recombination rates of charge carriers [21, 22]. To improve the electrical properties of binary CdSe and CdTe metal chalcogenides, we prepared ternary metal chalcogenide CdSe0.6Te0.4 composite photoelectrodes for PEC applications. Because the band gap energy of the CdSe and CdTe in between the solar spectrum (1-3 eV). To improve the electrical, optical and photoelectrical properties of the binary CdSe and CdTe thin films, we systemically de-signed experimental for the perfect composition of Se and Te ions with Cd like CdSe0.6Te0.4 thin films.”
Response to the Reviewer’s comments:
Thank you very much for your valuable comment. We agree to the option of the reviewer suggestions. We have proper correction has been done in the revised manuscript.
Comments 3.4] First, please check the grammar. For example, “Because the band gap energy of the CdSe and CdTe in between the solar spectrum (1-3 eV).” is not a sentence.
Response to the Reviewer’s comments:
Thank you very much for your valuable comment. The text is modified accordingly in the revised manuscript.
Comments 3.5] Second, it’s still not explained why CdSeTe is more beneficial or shows any additional properties.
Response to the Reviewer’s comments:
Thank you very much for your valuable comment. Mistakes were carefully identified and corrected in the revised manuscript.
Comments 3.6] Third, how do the authors define “a perfect composition”? This does not sound like a professional claim.
Response to the Reviewer’s comments:
Thank you very much for your valuable comment. We have mentioned the perfect composition, because previously researcher claimed that ternary CdSeXTe1-X compounds is the best composition of Se, and Te [1-3]. We agree to the reviewer comment and text has been revised.
[1] N. Muthukumarasamy, R. Balasundaraprabhu, S. Jayakumar, M. D. Kannan, P. Ramanathaswamy, Phys. Stat. Sol., 201, 2004, 2312–2318.
[2] S. Velumani, X. Mathew, P. J. Sebastian, Sol. Energ. Mater. Sol. C., 2003, 76, 359–368.
[3] R. Sathyamoorthy, P. Sudhagar, R. Saravana Kumar, P. Matheswaran, Ranjith G. Nair, Physica B, 2011, 406, 715–719
Round 3
Reviewer 3 Report
The authors have dealt with all the questions. I suggest acceptance.